# From Emergence to Evolution: Dynamics of the SARS-CoV-2 Omicron Variant in Florida

**DOI:** 10.3390/pathogens13121095

**Published:** 2024-12-12

**Authors:** Sobur Ali, Marta Giovanetti, Catherine Johnston, Verónica Urdaneta-Páez, Taj Azarian, Eleonora Cella

**Affiliations:** 1Burnett School of Biomedical Sciences, College of Medicine, University of Central Florida, Orlando, FL 32827, USA; mdsobur.ali@ucf.edu (S.A.); catherine.johnston@ucf.edu (C.J.); veronica.urdaneta@ucf.edu (V.U.-P.); 2Department of Sciences and Technologies for Sustainable Development and One Health, Università Campus Bio-Medico di Roma, 00128 Roma, Italy; giovanetti.marta@gmail.com; 3Oswaldo Cruz Institute, Oswaldo Cruz Foundation, Minas Gerais 30190-009, Brazil; 4Climate Amplified Diseases and Epidemics (CLIMADE)—CLIMADE Americas, Belo Horizonte 30190-002, Brazil

**Keywords:** SARS-CoV-2, genomic surveillance, molecular epidemiology, Florida, United States

## Abstract

The continual evolution of SARS-CoV-2 has significantly influenced the global response to the COVID-19 pandemic, with the emergence of highly transmissible and immune-evasive variants posing persistent challenges. The Omicron variant, first identified in November 2021, rapidly replaced the Delta variant, becoming the predominant strain worldwide. In Florida, Omicron was first detected in December 2021, leading to an unprecedented surge in cases that surpassed all prior waves, despite extensive vaccination efforts. This study investigates the molecular evolution and transmission dynamics of the Omicron lineages during Florida’s Omicron waves, supported by a robust dataset of over 1000 sequenced genomes. Through phylogenetic and phylodynamic analyses, we capture the rapid diversification of the Omicron lineages, identifying significant importation events, predominantly from California, Texas, and New York, and exportation to North America, Europe, and South America. Variants such as BA.1, BA.2, BA.4, and BA.5 exhibited distinct transmission patterns, with BA.2 showing the ability to reinfect individuals previously infected with BA.1. Despite the high transmissibility and immune evasion of the Omicron sub-lineages, the plateauing of cases by late 2022 suggests increasing population immunity from prior infection and vaccination. Our findings underscore the importance of continuous genomic surveillance in identifying variant introductions, mapping transmission pathways, and guiding public health interventions to mitigate current and future pandemic risks.

## 1. Introduction

Severe acute respiratory syndrome coronavirus 2 (SARS-CoV-2) has infected over 700 million people and caused more than seven million deaths globally since its initial detection in December 2019. The emergence of variants of concern (VOCs), characterized by increased transmissibility, disease severity, and immune evasion properties, has prolonged the COVID-19 pandemic, challenging global control efforts [1,2,3]. Among these, the Omicron variant, first reported in South Africa [2] in November 2021, spread globally at an unanticipated pace, swiftly replacing the Delta variant [3,4] and becoming the dominant lineage [1]. Omicron accumulated over 60 mutations in its genome, including 32 mutations in the spike protein, which were linked to increased transmissibility, infectivity, resistance to therapeutics, immune escapes, reinfection potential [5] and breakthrough infections. Omicron (B.1.1.529) diverged into five main lineages, designated BA.1 (and sub-lineage BA.1.1), BA.2 (and sub-lineage BA.2.12.1), BA.3, BA.4, and BA.5 [6]. Subsequent variants that have evolved from Omicron have continued to drive ongoing infections, highlighting its ability to persist and adapt over time.

Advancements in genome sequencing and phylogenetic methods have enabled large-scale analysis of SARS-CoV-2 genomic datasets, facilitating investigation of the viral evolution and the emergence of VOCs [7]. Multiple studies in the UK [8], Italy [9], Mexico [10], and South Korea [11] have been carried out to understand the origin and transmission dynamics of the Omicron variant. Together, they allow the comparison of viral transmission dynamics between geographies, providing insights into the global spread of the Omicron variants and the relative effectiveness of public health measures. A previous study combining over three million genomes and infection estimates in the US showed that Omicron variants infected around 30–47% of the US population [12]. Another recent study in the US reported the continuous emergence of new Omicron variants, like JN.1 and XBB, with immune escape substitution, highlighting the importance of continuous genomic surveillance [13]. Understanding the regional transmission dynamics of the Omicron variants, along with global and national efforts, is crucial. In particular, these data are important for pandemic preparedness efforts as well as the current response to SARS-CoV-2. Our study focuses on Florida, US, a major tourist and retirement destination known for its high elderly and immigrant populations. Omicron was first reported in Florida on December 7, 2021, and it spread rapidly across the state, leading to nearly double the daily COVID-19 cases reported during the Delta variant peak [14]. This rapid emergence posed challenges. Despite extensive vaccination efforts, Omicron’s swift spread underscored the need for genomic surveillance and proactive strategies to understand the viral dynamics and to inform future public health strategies.

Here, we investigate the SARS-CoV-2 viral evolution during the Omicron waves in Florida, expanding on prior research that examined the Delta wave in the state [15]. These data are important to our public health partners at the local, regional, and state levels, allowing for comparison to other jurisdictions. In particular, by analyzing the evolution, substitution patterns, and importation events of the Omicron lineages, this study provides critical insights into the factors driving the viral dynamics. A key strength of this work lies in its extensive genomic surveillance effort, which involved sequencing over 1000 genomes over two years. This robust dataset enables the exploration of the viral lineage expansion and displacement, shedding light on the intricate interplay between viral evolution, epidemic waves, and public health responses. Understanding these dynamics is crucial to developing more effective public health strategies and improving outbreak management. By identifying the primary drivers of high case numbers and mortality rates during the Omicron waves, this study delivers actionable insights for policymakers and healthcare authorities, supporting targeted interventions, mitigating future pandemic impacts, and ultimately enhancing public health outcomes.

## 2. Materials and Methods

### 2.1. Sample Collection

A total of 1814 positive swabs were collected from nasopharyngeal (NP) samples from college-aged individuals at a university student health service clinic (n = 1295) and a local children’s hospital (n = 519) in Orlando, FL between 1 April 2021 and 31 December 2022. Positive NP swabs were preserved in 2 mL DNA/RNA shield (Zymo Research, Irvine, CA, USA) and stored at 4 °C until RNA extraction.

### 2.2. RNA Extraction and RT-qPCR

Viral RNA was automatedly extracted from 140 μL of the sample using the QIAamp 96 virus QIAcube HT kit, following the manufacturer’s instructions. Fresh DNA/RNA shield was used as a negative control during the extraction. The RT-qPCR assays were performed in a 10 μL reaction volume, comprising 4X TaqPath master mix (Thermo Fisher Scientific, Waltham, MA, USA), qPCR probe 2019-nCoV_N1 from CDC (5′-FAM-ACCCCGCATTACGTTTGGTGGACC-BHQ1-3′), forward primer (5′-GACCCCAAAATCAGCGAAAT-3′), and reverse primer (5′-TCTGGTTACTGCCAGTTGAATCTG-3′) in a final concentration of 0.25 μM each, 4.25 μL of nuclease-free H_2_O, and 2.5 μL of extracted template RNA. The RT-qPCR amplification was conducted on a CFX Opus 96 instrument (Bio-Rad Laboratories, Hercules, CA, USA) under the following conditions: UNG (Uracil-N-Glycosylase) incubation at 25 °C for 2 min, a reverse transcription step at 50 °C for 15 min, followed by polymerase activation at 95 °C for 2 min, and finally, 35 cycles of amplification at 95 °C for 15 s and 55 °C for 30 s. All the samples were run in duplicate, including the positive control, negative extraction control, and no-template negative controls. Samples with a CT value below 29 were selected for sequencing [16].

### 2.3. Midnight Protocol for Oxford Nanopore Sequencing

SARS-CoV-2 genome sequencing was conducted using the Midnight RT-PCR expansion kit (EXP-MRT001) alongside the Rapid Barcoding Kit 96 (SQK-RBK110.96) protocol on the Oxford Nanopore sequencing platform. In brief, reverse transcription was performed to synthesize cDNA from the extracted viral RNA with the LunaScript RT master mix (New England Biolabs, Ipswich, MA, USA). Overlapping 1200 bp amplicons covering the SARS-CoV-2 genome were generated for each sample using the Midnight primer pools and Q5^®^ HS master mix (New England Biolabs). The resulting amplicons were pooled and barcoded with the Oxford Nanopore Rapid Barcoding Kit following the manufacturer’s instructions. After bead-based SPRI clean-up, the library was loaded onto an R9.4.1 flowcell and sequencing was conducted on a GridION X5 or MinION device using MinKNOW software. Real-time base-calling and demultiplexing were performed with the GridION software. Assembly was performed in two steps (using the default parameters) following the ARTIC Network bioinformatics protocol (https://artic.network/ncov-2019/ncov2019-bioinformatics-sop.html, accessed on 1 May 2021). Quality control and filtering (1000–1500 bp fragments) were applied using the gupplyplex script, and reads were assembled via the MinION pipeline, employing Medaka for variant-calling against the Wuhan-Hu-1 reference genome (GenBank accession MN908947.3). The lineages for each sample were assigned using the Pangolin tool.

### 2.4. Subsampling

We collected SARS-CoV-2 genome data from Florida and globally from January 2021 to December 2022 via GISAID to investigate the introduction and spread of the Omicron lineages in Florida. The Florida datasets were synchronized with globally representative sequences to accurately represent the viral introductions, minimizing the bias in ancestral state reconstruction due to sampling variability. Using US-specific and global subsampling techniques, the subsampler tool [17] enabled the random selection of sequences based on location, time, and epidemiological relevance. This approach required detailed sequence data and a matrix reflecting the case numbers for each variant of concern (VOC) throughout the study period. The case numbers were adjusted according to the estimated prevalence of each VOC in various countries, as documented in GISAID, with a baseline function guiding the sampling proportion. The resulting dataset, containing 4602 sequences, including 1119 newly sequenced genomes, encompassed all the major VOCs to illustrate the evolutionary relationships of our newly sequenced genomes. To specifically study the Omicron lineages in Florida, the final datasets comprised 18,942 BA.1 sequences (including 5469 from Florida), 17,846 BA.2 sequences (4448 from Florida), 14,540 BA.4 sequences (2936 from Florida), and 18,870 BA.5 sequences (4591 from Florida). Due to the limited available data, BA.3 sequences from Florida were excluded.

### 2.5. Phylogenetic Analyses

The sequences were initially aligned using ViralMSA with the default parameters [18]. Subsequently, manual curation was carried out using Aliview [19] to eliminate artifacts within the alignment and terminal regions, ensuring data integrity. Phylogenetic analysis was then conducted using the maximum likelihood (ML) method implemented in IQ-TREE v.2 [20]. The tree was inferred using the general time reversible (GTR) model of nucleotide substitution and a proportion of invariable sites (+I), determined by the ModelFinder application. The branch support was assessed via the approximate likelihood ratio test based on the bootstrap with 1000 replicates. To generate a time-scaled phylogeny, we used TimeTree with a constant evolutionary rate of 8.0 × 10^−4^ nucleotide substitutions per site per year to re-scale the branch lengths of the ML tree [21]. Outlier sequences that deviated from the strict molecular clock assumption, as identified by TimeTree, were systematically removed with the Ape package in R [22]. We then used TimeTree to estimate the number and source of viral introductions into Florida for each Omicron lineage. For this analysis, we fit a migration model for which the geographical locations of the taxon sampling were assigned to external (known) and internal nodes (inferred) [21]. Using a custom Python script, we counted the number of state changes by iterating over each phylogeny from the root to the external tips. We recorded the state changes whenever an internal node transitioned from one location to a different location in the resulting child node or tip(s). The timing of these transition events was then documented, serving as the estimated import or export events.

## 3. Results

Florida experienced five COVID-19 epidemic waves from 2020 to 2022. The first wave, registered in the early months of 2020, saw a rise in both cases and deaths, but it is not reported in this paper due to the lack of data for that year. While the total cases increased from 2021 to 2022, a significant reduction in the annual deaths was observed over the same period. The second wave, occurring from January to May 2021, was marked by the introduction and spread of the Alpha variant. This variant, with mutations conferring increased transmissibility and disease severity compared to the ancestral virus, led to a surge in daily cases and deaths. Following this wave, from May to June 2021, cases and deaths declined, marking a temporary easing of the pandemic (Figure 1a,b).

The third wave in Florida, driven by the Delta variant, lasted from 1 July to 31 October 2021. This wave resulted in a higher case count and increased mortality compared to the second wave. Daily cases began declining in September and continued to decrease through November 2021. Daily deaths peaked around 400 in September before gradually declining by December 2021 (Figure 1a,b). The Delta variant rapidly replaced the Alpha variant and remained dominant until November 2021. Despite the relatively high vaccination rate during this period, a significant proportion of the younger population remained unvaccinated. The relaxation of mask mandates and social-distancing measures, coupled with breakthrough infections—particularly in older adults and those with pre-existing conditions—likely contributed to the elevated mortality [23,24]. As reported for other countries and US regions [2,25,26,27], Florida’s fourth wave began in December 2021, driven by the emergence of the Omicron BA.1 variant (Figure 1a,b). This variant led to a dramatic spike in cases, with daily infections reaching approximatively 60,000, the highest recorded in Florida. By January 2022, Omicron BA.1 had swiftly displaced Delta, marking the start of the fourth major wave. Follow BA.1′s emergence, several Omicron sub-lineages were detected, including BA.2, BA.3, BA.4, and BA.5 (Figure 1).

BA.2 rapidly gained dominance by April 2022 due to its transmission advantage over BA.1, leading to Florida’s fifth wave. BA.2 also had the ability to reinfect individuals previously infected with BA.1, raising concerns about incomplete immunity [28,29,30,31,32]. As the BA.2 wave subsided in July 2022, BA.5 became the dominant strain, although this shift did not increase either cases or deaths. The epidemic plateaued with persistently high transmission levels and a gradual decline in cases into October 2022, likely due to increased herd immunity through vaccination and prior infection. In September 2022, a rapid shift to the BQ.1 variant, a sub-lineage of BA.5, was observed. Despite BQ.1′s potential to evade immune responses and designation as a variant of interest (VOI), its emergence did not lead to a new epidemic wave or result in a spike in cases or deaths (Figure 1).

We carried out genomic surveillance of the SARS-CoV-2 variants from samples collected between 1 April 2021 and 31 December 2022 across Orlando, a diverse and representative community in Central Florida. UCF’s large student population, proximity to major tourist attractions, and extensive transportation networks provided an ideal setting for analyzing the transmission dynamics. Of the 1814 samples collected, 1394 contained detectable viral RNA based on the RT-qPCR Ct values. We successfully sequenced 1119 SARS-CoV-2 genomes, which were publicly shared via GISAID to support global genomic surveillance efforts (Figure 1c,d).

Although our sample size for the Alpha variant was limited, we effectively captured the Delta variant’s emergence and dominance in Florida [19]. During the Omicron wave, we detected the BA.1 variant concurrent with its introduction, followed by the emergence of BA.2, BA.4, and BA.5 in alignment with their spread across Florida. Despite limited sampling during the later stages of the Omicron wave, we successfully identified the BQ.1 variant shortly after its introduction (Figure 1c,d). Vaccination played a crucial role throughout these waves (Figure 1e). While a significant portion of the population had completed their primary vaccine series by mid-2021, uptake of the booster doses lagged behind, particularly among younger populations [23,24]. The combination of the high transmissibility of new variants, the waning of vaccine-induced immunity, and delayed booster administration likely contributed to the continued spread of the virus during subsequent waves.

Further, we conducted a phylogenetic analysis by integrating our sequence data with additional sequences from Florida, the US, and global databases. This approach enabled a deeper understanding of the relationships and genetic distribution among the samples. Despite sampling from a geographically and demographically representative population, we did not observe any monophyletic clustering among our samples. Instead, the Florida genomes were dispersed across the phylogenetic tree, intermixed with sequences from various global regions (Figure 2).

We combined epidemiological data with phylodynamic analysis to explore the introduction and transmission dynamics of SARS-CoV-2 in Florida. Using ancestral state reconstruction on a time-calibrated phylogeny, we estimated the frequency of viral importation and exportation events between Florida and other locations (the continent and US states) (Figure 3). For the BA.1 lineage, we identified 826 international importation events, primarily from North America (82.4%) and Europe (16.7%), with the peak of introductions occurring around January 2022 (Figure 3a). Given the key role of North America in Florida’s transmission dynamics, we also focused on the transmission dynamics within the US. Among the US states, the highest number of BA.1 introductions originated from California (42.1%), followed by Texas (9.5%) and New York (7.4%) (Figure 3b). Florida also exported BA.1, predominantly to North America (75%), Europe (17.7%), and South America (6.1%). Nationally, the most frequent exportation destinations were California (19.7%), Texas (7.7%), and Colorado (7.8%) (Figure 3c). As suggested by Lopes [12], California and Texas, along with Florida, were among the US states with an higher daily peak in BA.1 infections.

We estimated 1630 imported events for the BA.2 lineage, primarily from North America (61.2%) and Europe (36.9%) (Figure 3d). The peak of these introductions occurred around May 2022, coinciding with a global surge in transmission [33]. Among the US states, most BA.2 introductions into Florida were traced to California (16.7%) and New York (12.4%). As a major travel hub, Florida became a key exporter of BA.2, with exportation primarily directed toward North America (66.9%) and Europe (23.3%). Within the US, Florida mainly exported BA.2 to California (15%) and New York (12.9%) (Figure 3d–f).

The Omicron BA.4 and BA.5 lineages, first reported in South Africa between January and February 2022 [7], subsequently spread more widely. The transmission dynamics of these lineages in Florida followed similar patterns (Figure 3g–l). We observed approximatively 1152 introductions of BA.4 and 1148 of BA.5 (Figure 3g,j). Most BA.4 and BA.5 importations originated from North America (90% and 84.5% for BA.4 and BA.5, respectively) and Europe (6.3% and 11.5%, respectively). Importation into Florida began increasing in March 2022 and peaked in June 2022. California contributed significantly to these introductions, accounting for 59% of BA.4 and 38.6% of BA.5 importations (Figure 3g,h,j,k). Regarding exportations, Florida mainly exported BA.4 and BA.5 to North America (70.6% and 50%, respectively), Europe (17.6% and 37.1%, respectively) and South America (11.2% for BA.4) (Figure 3g,j). Within the US, most export events were directed toward California (14% for BA.4, 21.2% for BA.5), Texas (9.6% for BA.4, 7.2% for BA.5) and New York (10.5% for BA.4 10% for BA.5) (Figure 3i,l).

## 4. Discussion

The rapid emergence of SARS-CoV-2 variants has posed a substantial challenge to global public health efforts, with the Omicron variant and its sub-lineages playing a pivotal role in shaping the COVID-19 pandemic’s trajectory. In our study, we employed genomic surveillance to investigate the importation and exportation dynamics of the Omicron variants in Florida, using samples collected from both a university and a hospital in Central Florida. Our sequencing efforts resulted in the successful sequencing and analysis of 1119 SARS-CoV-2 genomes, which contributed significantly to global genomic surveillance through platforms such as GISAID. It is noteworthy that the samples collected from these settings were representative of the overall pandemic in Florida. This is demonstrated by the alignment of the lineage distribution and proportion (Figure 1), which mirrors the broader trends observed throughout the pandemic [25,26,34]. This correspondence underscores the reliability and representativeness of the samples in capturing the dynamics of the pandemic in Florida. Our comprehensive analysis of the viral exchange has provided valuable insights into the dynamics of SARS-CoV-2 dissemination both into and out of Florida. The intricate relationship between international and domestic travel and viral transmission plays a crucial role in the emergence and spread of various SARS-CoV-2 variants. This underscores the necessity of continuous monitoring and coordinated global efforts to combat the spread of infectious diseases effectively. At the height of the BA.1 and BA.5 waves, Florida transitioned from being primarily a recipient of viral introductions to a major exporter of these lineages. Our results show that for BA.1 and BA.5, the number of exportation events from Florida was roughly double that of importations (Figure 3). This pattern highlights Florida’s role as a significant node in the national and international transmission networks, likely driven by its status as a major tourist destination and its large, mobile population. This viral migration pattern is consistent with the ones from other studies that synthesized viral genetic, epidemiological, and travel data [35]. Interestingly, our study also reveals distinct patterns for the BA.2 lineage, where Florida saw more importation than exportation events, with 1630 importations compared to 1267 exportations (Figure 3). This observation aligns with the ability of BA.2 to overcome immunity from prior Omicron BA.1 infection [36,37], which likely contributed to the high number of viral introductions into Florida during this period. Moreover, its global spread in early 2022 exhibited enhanced transmissibility and a higher potential for reinfection, as reported in other countries [9,38,39,40]. Our data further highlight the complex relationship between Florida and other US states in transmitting the SARS-CoV-2 variants. For the Omicron lineages, most importations came from key states such as California and New York, which are also major travel hubs. This finding is consistent with other studies that have demonstrated the importance of interstate travel and mobility in disseminating SARS-CoV-2 within the United States [12,41,42,43]. The number of exportations from Florida to states like California, Texas, and New York suggests that Florida played a crucial role in seeding outbreaks in other regions during the peak of the Omicron subvariant transmission. This was supported, as these states reported the highest daily BA.1 infections [12]. Such patterns of viral migration highlight the need for continued surveillance and coordination between states to track the movement of variants and implement timely public health interventions. Despite the high vaccination coverage in Florida during the study period (Figure 1), breakthrough infections were prevalent, particularly with BA.1 and BA.5. This is consistent with reports from similar studies, which suggest that the Omicron variants have a greater ability to evade vaccine-induced immunity [43,44,45,46]. This implies that the vaccine was more effective in reducing morbidity and mortality than the number of infections. Our results highlight the importance of booster vaccinations, as they have been shown to enhance protection against the Omicron variants, especially among vulnerable populations with comorbidities [47,48].

Our study possesses several limitations that should be acknowledged when evaluating the results. Although a significant quantity of SARS-CoV-2 sequences from Florida were analyzed, inconsistent data collected across regions may generate biases despite efforts to create a nationally representative dataset. The study of the geographical representation relied on some of the available sequences, which could have affected the accuracy of the conclusions drawn about how the virus spreads within the state. We did not measure air passenger traffic due to its free accessibility, which posed a limitation on our study. As a result, we can only speculate on the impact of air travel on the viral transmission dynamics. The lack of patient-specific metadata, including demographic information, vaccination status, and infection history, constrained our capacity to perform a more comprehensive epidemiological analysis. Despite these limitations, this study provides insights regarding SARS-CoV-2 transmission in Florida, underscoring the necessity of integrated initiatives in future studies.

In conclusion, our study provides important insights into the lineage replacement and the dynamics of SARS-CoV-2 importation and exportation in Florida during the Omicron waves. The transition from a net importer to an exporter of viral variants, particularly for BA.1 and BA.5, reflects broader global trends of variant spread. Moreover, the distinct patterns of BA.2 transmission emphasize the continued importance of genomic surveillance to monitor the introduction of new variants. These findings have significant implications for public health policy, particularly in relation to travel restrictions, vaccination campaigns, and the need for coordinated national and international surveillance efforts to curb the spread of emerging SARS-CoV-2 variants. Additionally, our findings highlight the complex dynamics of SARS-CoV-2 transmission and the pivotal role of both domestic and international travel in the spread of the virus. Continuous monitoring, global collaboration, and effective public health interventions are essential to mitigate the impact of current and future pandemics.

## Figures and Tables

**Figure 1 pathogens-13-01095-f001:**
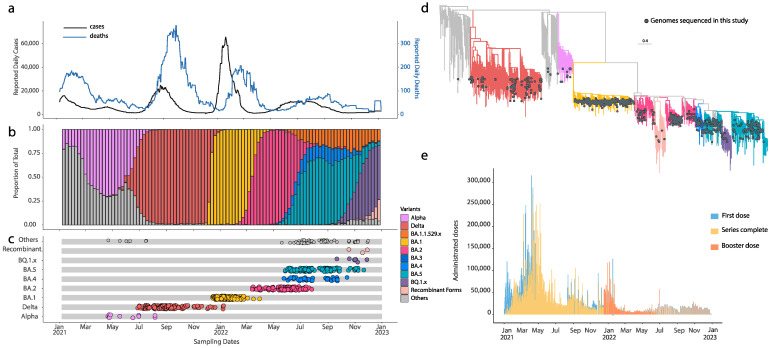
SARS-CoV-2 epidemic dynamics in Florida. (**a**) Daily count of COVID-19 cases (black) and associated deaths in Florida over time (light blue). (**b**) Progression in the proportion of sequenced SARS-CoV-2 isolates in Florida over the different waves of infection, showing the rapid replacement of different VOCs throughout time. The bars are colored-coded to represent the different variants according to the legend. (**c**) Temporal sampling of sequences in this study (n = 1119) through time with the VOCs highlighted and annotated according to their lineage assignment. (**d**) Time-resolved maximum likelihood phylogeny containing high-quality near-full genome sequences (n  =  1119) obtained from this study (colored in gray), analyzed against a backdrop of global reference sequences (n  =  3483). The branches are colored-coded according to the lineage assignment. (**e**) Vaccination impact across Florida over time. Colors highlight each vaccine dose.

**Figure 2 pathogens-13-01095-f002:**
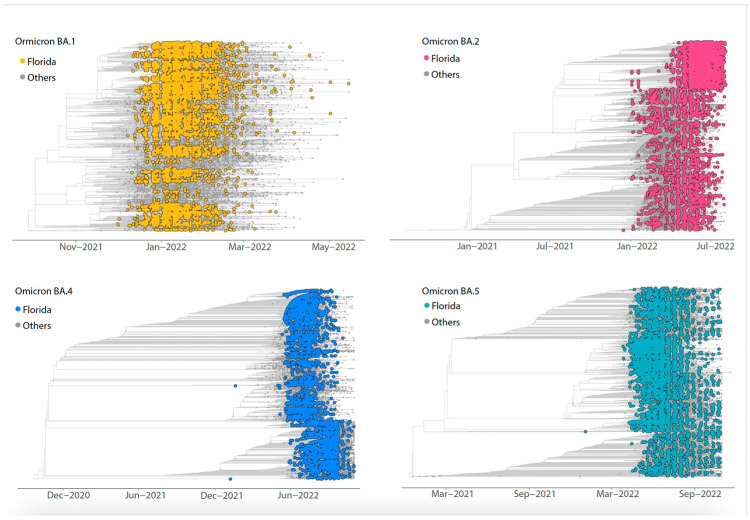
Phylogenetic distribution of Omicron lineages in Florida. Time-resolved maximum likelihood phylogenies of the Florida genomes analyzed against a backdrop of global reference sequences for each Omicron lineage (BA.1, BA.2, BA.4 and BA.5). The tips of the Florida genomes are colored according to the legend on the top left.

**Figure 3 pathogens-13-01095-f003:**
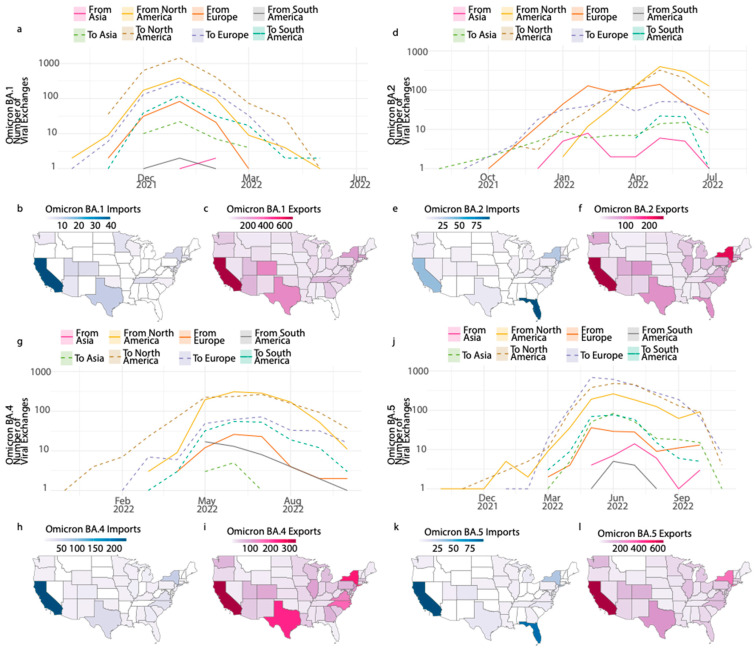
Inferred viral dissemination patterns of Omicron lineages (BA.1, BA.2, BA.4 and BA.5) in Florida. Inferred viral exchange patterns to and from Florida for the Omicron lineages (**a**,**d**,**g**,**j**). Introductions in Florida are shown in solid lines and exports from Florida are shown in dotted lines and these are colored by continent. Inferred locations of importations (**b**,**e**,**h**,**k**) and exportations (**c**,**f**,**i**,**l**) of the Omicron lineages that drove the Florida epidemic over time.

## Data Availability

All genome sequences from Florida and the other countries used in this study are freely available in the GISAID databases. In addition, the dataset used in this study is available upon request from the corresponding authors.

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
