# Peer review of "From Emergence to Evolution: Dynamics of the SARS-CoV-2 Omicron Variant in Florida"

_pathogens, 2024, doi:10.3390/pathogens13121095_

Round 1
Reviewer 1 Report
Comments and Suggestions for Authors
The authors highlight the impact of SARS-CoV-2 evolution on the COVID-19 pandemic, focusing on the Omicron variant, which emerged in late 2021 and became the dominant strain globally due to its high transmissibility and immune evasion. In Florida, Omicron led to unprecedented case surges despite vaccination efforts. This study investigates Omicron's molecular evolution and transmission dynamics in Florida, analyzing over 1,000 genomes to explore lineage diversification, mutations, and import/export patterns. Key findings include significant variant importation from California, Texas, and New York, and exportation to North America and Europe. The authors emphasize the importance of genomic surveillance in informing public health strategies. After carefully reviewing this manuscript, I believe the manuscript cannot be accepted in this current form. Furthermore, the study on Omicron might be a bit outdated, and not important in the current setting, hence authors might need to add some paragraphs to support the importance of this work
1. Authors need to increase the font size for all the figures and also proofread the whole document for typos.
2. Lines 20–22, lines 23–25, etc., can be streamlined. The abstract needs to focus on other key findings rather than repeatedly highlighting the role of genomic surveillance and tracking variant evolution. While these are critical points, their recurrence detracts from the abstract’s focus and compresses space for other key findings or novel insights.
3. The paper focuses on a major event that has already taken place and is no longer important. If it is still important, authors need to justify its relevance.
4. Why did the authors not fully investigate the role of international travel since Florida is a major tourism resort?
5. Why did the authors focus on Omicron in Florida? This wasn’t well justified. Many authors have already studied this topic, i.e. the biology. See:
Lopes R, Pham K, Klaassen F, Chitwood MH, Hahn AM, Redmond S, Swartwood NA, Salomon JA, Menzies NA, Cohen T, Grubaugh ND. Combining genomic data and infection estimates to characterize the complex dynamics of SARS-CoV-2 Omicron variants in the US. Cell Rep. 2024 Jul 23;43(7):114451. doi: 10.1016/j.celrep.2024.114451. Epub 2024 Jul 5. PMID: 38970788.
Parsons RJ, Acharya P. Evolution of the SARS-CoV-2 Omicron spike. Cell Rep. 2023 Dec 26;42(12):113444. doi: 10.1016/j.celrep.2023.113444. Epub 2023 Nov 18. PMID: 37979169; PMCID: PMC10782855.
Liu W, Huang Z, Xiao J, Wu Y, Xia N, Yuan Q. Evolution of the SARS-CoV-2 Omicron Variants: Genetic Impact on Viral Fitness. Viruses. 2024 Jan 25;16(2):184. doi:
10.3390/v16020184. PMID: 38399960; PMCID: PMC10893260.
Etc
6. Furthermore, the introduction lacks a clear explanation of the study’s novelty compared to existing literature. That needs to be clearly outlined
7. Did the public health measures (like the mask mandates) affect the observed dynamics in Florida. What impact did they play
8. Line 316: Ability of BA.2 to evade immunity? Authors need to cite this.
9. Since Omicron was an international virus/problem, why did the authors not compare Florida's dynamics with other states or global regions?
10. While vaccination boosters have been mentioned, it was not thoroughly analyzed, hence leaving the discussion on breakthrough infections incomplete.
11. The authors need to give enough consideration to the biological significance of the identified mutations in this study.
12. Rewrite the conclusion to provide a comprehensive discussion of the study's findings, currently it is lacking. Clearly interpret the results, address limitations, and provide suggestions for future research. E.g. statistical limitations and sampling methods need to be highlighted, also The conclusion must provide some recommendations to policymakers arising from this study for now and future studies.
Author Response
The authors highlight the impact of SARS-CoV-2 evolution on the COVID-19 pandemic, focusing on the Omicron variant, which emerged in late 2021 and became the dominant strain globally due to its high transmissibility and immune evasion. In Florida, Omicron led to unprecedented case surges despite vaccination efforts. This study investigates Omicron's molecular evolution and transmission dynamics in Florida, analyzing over 1,000 genomes to explore lineage diversification, mutations, and import/export patterns. Key findings include significant variant importation from California, Texas, and New York, and exportation to North America and Europe. The authors emphasize the importance of genomic surveillance in informing public health strategies. After carefully reviewing this manuscript, I believe the manuscript cannot be accepted in this current form. Furthermore, the study on Omicron might be a bit outdated, and not important in the current setting, hence authors might need to add some paragraphs to support the importance of this work.
Reply: We thank the reviewer for the constructive feedback. We have carefully revised the manuscript to address your suggestions to emphasize the importance of continued investigations of the SARS-CoV-2 pandemic.
- Authors need to increase the font size for all the figures and also proofread the whole document for typos.
Reply: Thank you for the feedback. We have increased the font size in the figures to improve readability. We have proofread the manuscript.
- Lines 20–22, lines 23–25, etc., can be streamlined. The abstract needs to focus on other key findings rather than repeatedly highlighting the role of genomic surveillance and tracking variant evolution. While these are critical points, their recurrence detracts from the abstract’s focus and compresses space for other key findings or novel insights.
Reply: We thank the reviewer for bringing this to our attention. We have now made the necessary changes.
- The paper focuses on a major event that has already taken place and is no longer important. If it is still important, authors need to justify its relevance.
Reply: Thank you to the reviewer for bringing this to our attention. Certainly, we feel that thorough investigation of the pandemic is important for future public health preparedness activities, as is the investigation of any historical epidemic. A global focus of SARD-CoV-2 transmission dynamics often obscures local dynamics at a sub-country level. Therefore we believe that these targeted analysis are important to juxtapose regional and global population processes. We have now better emphasized the importance of our work in the manuscript.
- Why did the authors not fully investigate the role of international travel since Florida is a major tourism resort?
Reply: We have investigated both local and national sources for viral transmission dynamics. Given that the primary source and destination was North America, we further focused our investigation within US states. Unfortunately, air travel data was not readily available for analysis, and not the primary focus of the study.
- Why did the authors focus on Omicron in Florida? This wasn’t well justified. Many authors have already studied this topic, i.e. the biology. See:
Lopes R, Pham K, Klaassen F, Chitwood MH, Hahn AM, Redmond S, Swartwood NA, Salomon JA, Menzies NA, Cohen T, Grubaugh ND. Combining genomic data and infection estimates to characterize the complex dynamics of SARS-CoV-2 Omicron variants in the US. Cell Rep. 2024 Jul 23;43(7):114451. doi: 10.1016/j.celrep.2024.114451. Epub 2024 Jul 5. PMID: 38970788.
Parsons RJ, Acharya P. Evolution of the SARS-CoV-2 Omicron spike. Cell Rep. 2023 Dec 26;42(12):113444. doi: 10.1016/j.celrep.2023.113444. Epub 2023 Nov 18. PMID: 37979169; PMCID: PMC10782855.
Liu W, Huang Z, Xiao J, Wu Y, Xia N, Yuan Q. Evolution of the SARS-CoV-2 Omicron Variants: Genetic Impact on Viral Fitness. Viruses. 2024 Jan 25;16(2):184. doi:
10.3390/v16020184. PMID: 38399960; PMCID: PMC10893260.
Reply: We thank the reviewer for raising this point. Here, our focus was twofold: (i) conducting an extensive genomic surveillance effort, which involved sequencing over 1,000 genomes over two years, and (ii) utilizing that dataset in combination with other genomes sampled from Florida cases to explore viral lineage expansion and replacement. This approach elucidated the viral evolution, epidemic waves, and public health responses in Florida. While Lopes et al. conducted an impressive analysis integrating epidemiological models with genomic data to study Omicron dynamics in the US, other studies have focused more on viral fitness and/or spike mutations. Further, local dynamics are often lost in global or national-level analysis. Therefore, we believe a focused investigation at the state level is warranted. In particular, these data can be used by local and state public health officials for pandemic preparedness.
- Furthermore, the introduction lacks a clear explanation of the study’s novelty compared to existing literature. That needs to be clearly outlined
Reply: Thank you for bringing this to our attention. We have now included relevant information in the introduction to emphasize the importance of our work. While our approach and methods may not be novel, we appropriately apply them to investigate an important component of local transmission dynamics. As omicron was such an epidemiologically important variant, documenting its emergence and replacement of the previous strain is relevant to understanding broader viral transmission dynamics.
- Did the public health measures (like the mask mandates) affect the observed dynamics in Florida. What impact did they play
Reply: Thank you for your question. The study covered a period beginning in 2021 when Florida had already lifted most state-level public health mandates, including mask requirements, and had a predominantly open economy. Among the 67 counties, some implemented their own measures; however, these were not systematically tracked. These policy decisions created a unique setting to study SARS-CoV-2 dynamics in a largely unmitigated environment. The absence of stringent public health measures likely contributed to the rapid and widespread transmission of the Omicron variant in the state.
- Line 316: Ability of BA.2 to evade immunity? Authors need to cite this.
Reply: We thank the reviewer for the comment. We have now included the appropriate citations.
- Since Omicron was an international virus/problem, why did the authors not compare Florida's dynamics with other states or global regions?
Reply: Thank you for your insightful suggestion. We acknowledge the importance of comparing Florida's dynamics with those of other states or global regions to provide a broader context. As they noted previously, a national and global investigations of SARS-CoV-2 Omicron population dynamics have previously been published. We sought to focus on Florida due to its unique characteristics—its high elderly population, significant tourism hub, immigration rates, and relaxed public health mandates, which influence viral transmission and public health responses differently than in other regions. Further, as researchers based in Florida, these data are important to our public health partners at the county and state level. While global and interstate comparisons are indeed valuable, this study's scope is to provide a detailed, localized analysis of Florida’s dynamics. This can serve as a foundation for future comparative studies and contribute to understanding the impact of differing public health measures across regions. We aim to address Florida-specific challenges and inform targeted interventions within the state.
- While vaccination boosters have been mentioned, it was not thoroughly analyzed, hence leaving the discussion on breakthrough infections incomplete.
Reply: Thank you for bringing this to our attention. We improve the discussion and expanded on the topic of breakthrough infections.
- The authors need to give enough consideration to the biological significance of the identified mutations in this study.
Reply: We thank the reviewer for their suggestion. However, our analysis of the mutational profile of our sample did not reveal any novel mutations compared to the well-known Omicron variants.
- Rewrite the conclusion to provide a comprehensive discussion of the study's findings, currently it is lacking. Clearly interpret the results, address limitations, and provide suggestions for future research. E.g. statistical limitations and sampling methods need to be highlighted, also The conclusion must provide some recommendations to policymakers arising from this study for now and future studies.
Reply: We have now redrafted the conclusion to better address the main key points. In particular , we refocused our discussion on the ability of Omicron as a vaccine breakthrough. We included relevant studies on both international and national viral spread. In the conclusion, we provide recommendations to policymakers based on this study for current and future considerations. As well as limitations of the study.

Reviewer 2 Report
Comments and Suggestions for Authors
Report on “From Emergence to Evolution: Dynamics of the SARS-CoV-2 Omicron Variant in Florida.”
Ref: Pathogens-3346266
Title: From Emergence to Evolution: Dynamics of the SARS-CoV-2 Omicron Variant in Florida.
Journal: Pathogens
In this article, the author investigated the molecular evolution and transmission dynamics of SARS-CoV-2 Omicron lineages during the Omicron waves in Florida. Furthermore, the authors
analyzed lineage diversification, mutations, and the role of importation and exportation events by leveraging a genomic dataset, including the sequencing of over 1,000 genomes across the state. The authors examined how Omicron lineages have evolved and displaced prior variants, providing a detailed overview of viral introduction and transmission patterns. This study provides insights into the drivers of variant dominance, including key environmental and demographic factors. The authors deduced a significant importation of the Omicron variant in Florida from California, Texas, and New York and exportation to North American and European regions. Furthermore, the investigation underscored the crucial role of genomic surveillance in identifying emerging variants and guiding targeted public health interventions to mitigate ongoing and future pandemic threats.
In my opinion, the topic is interesting, the sample collection, the RNA extraction and RT-qPCR, the phylogenetic analyses, and the concluding discussions presented in this paper are interesting.
Some minor remarks:
1. The introduction needs to include more details about previous related studies regarding molecular evolution and transmission dynamics of SARS-CoV-2 Omicron lineages in other regions or countries. The authors should highlight the novelty of their work when compared with similar studies for other regions or countries.
2. The authors are encouraged to add some sentences on how their study can provide the multiple origins of Omicron in Florida and its subsequent spread, both domestically and internationally.
3. How can this study offer valuable insights into the transnational transmission patterns and adaptive evolution of Omicron?
4. The English needs to be slightly checked since there are some problems with some sentence structures.
5. I suggest adding the following reference related to this work and discussing them inside the text:
§ Corbeil A, Johnstone J, Macdonald L, Schwartz KL, Bruce Barrett C, Reinprecht J, Heisey A, Nasso S, Jüni P, Campigotto A. Viral Dynamics of the SARS-CoV-2 Omicron Variant in Pediatric Patients: A Prospective Cohort Study. Clin Infect Dis. 2024;78(6):1506-1513. https://doi.org/10.1093/cid/ciad740
§ Eales, O., de Oliveira Martins, L., Page, A.J. et al. Dynamics of competing SARS-CoV-2 variants during the Omicron epidemic in England. Nat Commun 13, 4375 (2022). https://doi.org/10.1038/s41467-022-32096-4
§ Razzaq A, Disoma C, Iqbal S, Nisar A, Hameed M, Qadeer A, Waqar M, Mehmood SA, Gao L, Khan S and Xia Z (2024) Genomic epidemiology and evolutionary dynamics of the Omicron variant of SARS-CoV-2 during the fifth wave of COVID-19 in Pakistan. Front. Cell. Infect. Microbiol. 14:1484637. https://doi.org/10.3389/fcimb.2024.1484637
Author Response
Report on “From Emergence to Evolution: Dynamics of the SARS-CoV-2 Omicron Variant in Florida.”
Ref: Pathogens-3346266
Title: From Emergence to Evolution: Dynamics of the SARS-CoV-2 Omicron Variant in Florida.
Journal: Pathogens
In this article, the author investigated the molecular evolution and transmission dynamics of SARS-CoV-2 Omicron lineages during the Omicron waves in Florida. Furthermore, the authors analyzed lineage diversification, mutations, and the role of importation and exportation events by leveraging a genomic dataset, including the sequencing of over 1,000 genomes across the state. The authors examined how Omicron lineages have evolved and displaced prior variants, providing a detailed overview of viral introduction and transmission patterns. This study provides insights into the drivers of variant dominance, including key environmental and demographic factors. The authors deduced a significant importation of the Omicron variant in Florida from California, Texas, and New York and exportation to North American and European regions. Furthermore, the investigation underscored the crucial role of genomic surveillance in identifying emerging variants and guiding targeted public health interventions to mitigate ongoing and future pandemic threats.
In my opinion, the topic is interesting, the sample collection, the RNA extraction and RT-qPCR, the phylogenetic analyses, and the concluding discussions presented in this paper are interesting.
Reply: Thank you very much for your thoughtful and constructive feedback.
Some minor remarks:
- The introduction needs to include more details about previous related studies regarding molecular evolution and transmission dynamics of SARS-CoV-2 Omicron lineages in other regions or countries. The authors should highlight the novelty of their work when compared with similar studies for other regions or countries.
Reply: Thank you for bringing this to our attention. We have now included relevant information in the introduction to emphasize the novelty of our work.
- The authors are encouraged to add some sentences on how their study can provide the multiple origins of Omicron in Florida and its subsequent spread, both domestically and internationally.
Reply: We have now added more sentences to discuss the Omicron spread in Florida. In particular see lines 428-432; 438-439; 446-452.
- How can this study offer valuable insights into the transnational transmission patterns and adaptive evolution of Omicron?
Reply: Thank you for your comment. This work provides considerable insights into the transmission dynamics and adaptive evolution of Omicron in the context of several critical factors. Through the analysis of more than 1,000 SARS-CoV-2 genomes from Florida, alongside publicly available worldwide genomic data, we observed importation events and interconnections across viral lineages from other areas. This approach reveals the routes of global viral spread and underscores Florida's role in Omicron transmission, attributable to its high tourist and migration rates. Furthermore, our analysis of lineage evolution during the Omicron waves enhances our comprehension of the virus's adaptation to selection forces, including vaccination and pre-existing immunity. These data are important to our local and state public health, who may use them for pandemic planning and preparedness activities.
- The English needs to be slightly checked since there are some problems with some sentence structures.
Reply: We have carried out a thorough proofreading of the manuscript and addressed any inconsistencies with the text.
- I suggest adding the following reference related to this work and discussing them inside the text:
- Corbeil A, Johnstone J, Macdonald L, Schwartz KL, Bruce Barrett C, Reinprecht J, Heisey A, Nasso S, Jüni P, Campigotto A. Viral Dynamics of the SARS-CoV-2 Omicron Variant in Pediatric Patients: A Prospective Cohort Study. Clin Infect Dis. 2024;78(6):1506-1513. https://doi.org/10.1093/cid/ciad740
- Eales, O., de Oliveira Martins, L., Page, A.J. et al. Dynamics of competing SARS-CoV-2 variants during the Omicron epidemic in England. Nat Commun 13, 4375 (2022). https://doi.org/10.1038/s41467-022-32096-4
- Razzaq A, Disoma C, Iqbal S, Nisar A, Hameed M, Qadeer A, Waqar M, Mehmood SA, Gao L, Khan S and Xia Z (2024) Genomic epidemiology and evolutionary dynamics of the Omicron variant of SARS-CoV-2 during the fifth wave of COVID-19 in Pakistan. Front. Cell. Infect. Microbiol. 14:1484637. https://doi.org/10.3389/fcimb.2024.1484637
Reply: We appreciate the suggestion. We have now included the relevant references.

Round 2
Reviewer 1 Report
Comments and Suggestions for Authors
The manuscript needs to be proofread for typos and sentence continuity; otherwise, it now looks good.
Author Response
The manuscript needs to be proofread for typos and sentence continuity; otherwise, it now looks good.
Reply: We thank the reviewer for the constructive feedback. We have thoroughly proofread the manuscript and addressed any inconsistencies with the text.